# Exposure to Stress and Air Pollution from Bushfires during Pregnancy: Could Epigenetic Changes Explain Effects on the Offspring?

**DOI:** 10.3390/ijerph18147465

**Published:** 2021-07-13

**Authors:** Vanessa E. Murphy, Wilfried Karmaus, Joerg Mattes, Bronwyn K. Brew, Adam Collison, Elizabeth Holliday, Megan E. Jensen, Geoffrey G. Morgan, Graeme R. Zosky, Vanessa M. McDonald, Edward Jegasothy, Paul D. Robinson, Peter G. Gibson

**Affiliations:** 1Priority Research Centre Grow Up Well, Hunter Medical Research Institute, University of Newcastle, Newcastle, NSW 2308, Australia; joerg.mattes@newcastle.edu.au (J.M.); adam.collison@newcastle.edu.au (A.C.); megan.jensen@newcastle.edu.au (M.E.J.); 2Priority Research Centre for Healthy Lungs, Hunter Medical Research Institute, University of Newcastle, Newcastle, NSW 2308, Australia; vanessa.mcdonald@newcastle.edu.au (V.M.M.); peter.gibson@health.nsw.gov.au (P.G.G.); 3Faculty of Health, School of Medicine and Public Health, University of Newcastle, Newcastle, NSW 2308, Australia; liz.holliday@newcastle.edu.au; 4Division of Epidemiology, Biostatistics, and Environmental Health Science, School of Public Health, The University of Memphis, Memphis, TN 38152, USA; karmaus1@memphis.edu; 5Paediatric Respiratory and Sleep Medicine Department, John Hunter Children’s Hospital, Newcastle, NSW 2305, Australia; 6National Perinatal Epidemiology and Biostatistics Unit, Centre for Big Data Research in Health, Department of Medicine, School of Women’s and Children’s Health, University of New South Wales, Sydney, NSW 2052, Australia; b.haasdyk@unsw.edu.au; 7University Centre for Rural Health, Sydney School of Public Health, University of Sydney, Sydney, NSW 2006, Australia; geoffrey.morgan@sydney.edu.au (G.G.M.); edward.jegasothy@sydney.edu.au (E.J.); 8Menzies Institute for Medical Research, University of Tasmania, Hobart, TAS 7000, Australia; graeme.zosky@utas.edu.au; 9Tasmanian School of Medicine, University of Tasmania, Hobart, TAS 7000, Australia; 10School of Nursing and Midwifery, University of Newcastle, Newcastle, NSW 2308, Australia; 11Department of Respiratory Medicine, The Children’s Hospital at Westmead, Sydney, NSW 2145, Australia; paul.robinson1@health.nsw.gov.au; 12Discipline of Paediatrics and Child Health, University of Sydney, Sydney, NSW 2006, Australia; 13Department of Respiratory and Sleep Medicine, John Hunter Hospital, Newcastle, NSW 2305, Australia

**Keywords:** wildfire, epigenetics, DNA methylation, PM2.5, asthma, pregnancy

## Abstract

Due to climate change, bushfires are becoming a more frequent and more severe phenomenon which contributes to poor health effects associated with air pollution. In pregnancy, environmental exposures can have lifelong consequences for the fetus, but little is known about these consequences in the context of bushfire smoke exposure. In this review we summarise the current knowledge in this area, and propose a potential mechanism linking bushfire smoke exposure in utero to poor perinatal and respiratory outcomes in the offspring. Bushfire smoke exposure is associated with poor pregnancy outcomes including reduced birth weight and an increased risk of prematurity. Some publications have outlined the adverse health effects on young children, particularly in relation to emergency department presentations and hospital admissions for respiratory problems, but there are no studies in children who were exposed to bushfire smoke in utero. Prenatal stress is likely to occur as a result of catastrophic bushfire events, and stress is known to be associated with poor perinatal and respiratory outcomes. Changes to DNA methylation are potential epigenetic mechanisms linking both smoke particulate exposure and prenatal stress to poor childhood respiratory health outcomes. More research is needed in large pregnancy cohorts exposed to bushfire events to explore this further, and to design appropriate mitigation interventions, in this area of global public health importance.

## 1. Introduction

Major bushfire events are increasing due to global climate change, both in number and severity of events, which contributes to poor air quality and associated health problems. Environmental exposures, such as bushfire smoke, during pregnancy have the potential to alter health and disease in the offspring, however the mechanisms for these effects are not established. This knowledge is crucial if we are to mitigate the effects of these exposures.

Exposure to air pollutants, tobacco smoke, and maternal stress are known to influence the in utero environment and are associated with adverse outcomes at birth and in later life [1,2]. A key mechanism by which early environmental exposures could influence health outcomes is via epigenetics. Epigenetic changes modify DNA structure without changing the DNA sequence and allow regulation of gene expression and facilitate alternate splicing [3], allowing the body to respond to a changed environment without requiring long-term adaptation of the genome. DNA methylation (DNAm) of cytosine in the cytosine-phosphate-guanine (CpG) site of gene promoter regions can block the activity of a gene promoter, thus suppressing gene expression. Alternatively, methylation of CpG sites in the gene body can modify splicing, resulting in production of different messenger RNAs (mRNA). This increases the diversity of transcripts and proteins and can result in phenotypic changes, including increased vulnerability to disease.

Recent bushfire crises in Australia and other parts of the world bring into question the effect of bushfire smoke on vulnerable population groups, including pregnant women and their infants. The Australian bushfire events of 2019/2020 resulted in extreme smoke exposure in Eastern Australia, with heightened levels of PM2.5 (fine particulate matter <2.5 µm in size) [4] and stress among pregnant women with asthma (Asthma Australia, personal communication) [5]. Both stress and air pollutant exposure during pregnancy have been linked to poor perinatal outcomes, such as preterm birth, and the development of asthma, via epigenetic mechanisms [2,6,7]. However, most existing data on air pollutants, for example, relates to chronic lower level exposures in urban settings from a wide range of sources including motor vehicle emissions, industrial emissions and emissions from fossil fuel power generation. Bushfire smoke exposure differs from this, being a high concentration exposure, and, in the case of the recent bushfire events in Australia, also a prolonged exposure, over weeks and months. There is also a lack of information on whether bushfire smoke exposure can change the epigenome, thus producing long-term health risks, particularly after in utero exposure. This review will summarise the existing literature regarding perinatal health effects of intra-uterine bushfire smoke exposure and associations with DNAm.

The increasingly important issue of the effects of bushfire (wildfire) smoke exposure on human health is highly topical, and has important public health implications both in Australia and globally. The 2019/2020 Eastern Australian bushfires were of unprecedented scale both in terms of duration (>133-day exposure) and intensity. A survey completed by 7285 people with asthma and 4867 people without asthma at the height of the fire event (December 2019–January 2020) demonstrated the substantial acute impact of the crisis particularly on people with asthma [5]. Children with asthma were more likely to have emergency department (ED) visits or hospital admissions as a result of exposure to bushfire smoke [5]. The significant, immediate health burden of this prolonged fire activity included an estimated 417 (95% confidence interval [CI] 153–680) excess deaths, and 1305 (95% CI 705–1908) additional ED presentations for asthma due to bushfire smoke exposure [4]. Levels of PM2.5 were >95th percentile of historical daily averages (for at least 1 air quality monitor) on 94% of days between 1 October 2019 and 10 February 2020, and were above the national air quality standard for 24-h PM2.5 (25 µg/m^3^) on at least 22 days, with a maximum PM2.5 of 98.5 µg/m^3^ on 14 January [4]. A recent review called for future research in “sub-populations with heightened vulnerability” [8]. Infants of pregnant women with asthma, who are at particularly high risk of developing asthma themselves, are one such group which has yet to be investigated.

### 1.1. Effects of Bushfire Smoke Exposure on Pregnancy Outcomes

Previous studies have demonstrated that exposure to bushfire smoke during pregnancy is associated with an increased risk of adverse pregnancy outcomes, but none have investigated the mechanisms involved. A population cohort study of the 2009 Black Saturday bushfires in Victoria, Australia, found that exposure to the fires in the second and third trimester was associated with more preterm births and lower birth weight [9]. The fifteen fires during this period were collectively known as the Black Saturday fires and burned for 31 days up to 9 March 2009. Data from all births in Victoria between February 2009 and October 2009 were compared between areas which were fire-affected (>17,000 births), and those which were unaffected (>55,000 births), and the same analysis was conducted for births from 2006 to 2008 to control for any pre-existing differences between these regions. In babies born between February to April 2009 (mothers exposed later in pregnancy), there was a 50% increase in the proportion born between 20–27 weeks gestation, a 150% increase in babies <500 g, and a significantly lower mean birth weight among those in fire-affected areas compared to non-fire-affected areas. There were smaller changes observed when mothers had been exposed earlier in pregnancy, such as significantly fewer term births, but no differences in birth weights were found. The authors of this study discussed the potential effects of “bushfire stress” on pregnancy [9]; however, this study did not consider the potential impact of smoke exposure itself.

A study of the 2003 Canberra bushfires in Australia compared birth weight and gestational age among all births in the region in the year of the fire (2003) with the three previous years (2000–2002), and the seven years following (2004–2010) [10]. Exposure was categorised as “severely affected” (areas where death and property damage occurred), “moderately affected” (only property damage), and “least affected” (the remainder). There was no association between fire exposure and gestational age; however, birth weights were higher in the severely affected area, particularly among male infants. The authors hypothesise that stress-related increases in cortisol may have contributed to further increases in blood glucose levels, increasing the risk of macrosomia [10].

In contrast, data from the 2003 Southern California Wildfires [11] (21-day exposure) and wildfire smoke events in Colorado between 2007 and 2015 [12] (intermittent exposure), both found an association between fire smoke exposure during pregnancy and reduced birth weight. In California, offspring of women exposed during the third trimester had a significantly lower birth weight at term, on average 7.0 g lower than from unexposed women (95% CI −11.8 g, −2.2 g), while offspring exposed in the second trimester had a significantly lower birth weight of 9.7 g (95% CI −14.5 g, −4.8 g) [11]. The estimate of PM2.5 exposure was 90 µg/m^3^ under heavy smoke conditions and 75 µg/m^3^ under light smoke conditions, compared to 20 µg/m^3^ under normal conditions [13].

The Colorado study examined the impact of wildfire smoke from upwind regions (including California), which is a major source of ambient PM2.5, on adverse pregnancy outcomes [12]. In this study, all singleton births with a gestational age between 30 and 42 weeks from 2007 to 2015 were included and a range of perinatal outcomes examined. Exposure to PM2.5 (either from wildfire smoke, or non-smoke related) was characterised using ZIP codes and satellite imagery to determine smoke plume extent, as well as ground based PM2.5 monitors. Due to the intermittent nature of wildfire smoke events in Colorado, the PM2.5 concentrations attributed to wildfire smoke were low with a maximum of 4.5 µg/m^3^. However, for each 1 µg/m^3^ increase in average first trimester wildfire smoke PM2.5 exposure, there was a significant 5.7 g decrease in birth weight, which was not found for the second or third trimesters. For each 1 µg/m^3^ increase in exposure to wildfire smoke PM2.5 over the entire pregnancy, there was a significant 7.6% increased odds of preterm birth (after adjusting for maternal age, smoking, asthma, socioeconomic status, other environmental exposures such as ozone and PM10 [particulate matter <10 µm in size], and time of year). In mothers, adverse associations were also observed between wildfire smoke exposure and gestational diabetes and pregnancy induced hypertension [12].

Studies of bushfire events in Australia and globally have consistently demonstrated effects on fetal growth and an increase in the proportion of babies born prematurely. A systematic review of 11 studies which assessed the impact of maternal exposure to acute air pollution events (including four studies focused on wildfires), found that the most consistent finding was an association with reduced birth weight and increased preterm birth, while other obstetric complications were not often reported [14]. Being born early or small constitutes a risk for the child’s future health, such as an increased risk of cardiovascular, metabolic, and respiratory diseases later in life.

In pregnancy, smoke exposure may lead to hypoxia, or increased oxidative stress at the maternal–fetal interface. In the case of pregnant women with asthma, acute effects of the bushfire smoke exposure may directly affect asthma, increasing inflammation, reducing lung function, or increasing the risk of asthma exacerbation, which is known to be associated with poor perinatal outcomes in this specific population [15,16].

The Hazelwood Health Study has been investigating the health effects of another acute high concentration smoke exposure event, the Hazelwood coal mine fire in Victoria, Australia [17]. This fire burned for 51 days in February and March 2014, and covered towns in regional Victoria in plumes of smoke and ash. In this study, 571 children who were exposed either in utero, up to 2 years of age, or conceived after the fires, continue to be followed-up, while de-identified data from 3679 children born between March 2012 and December 2015 was used to assess effects on child health outcomes [17]. Average PM2.5 exposure attributable to the fire over the duration of pregnancy was 4.4 µg/m^3^ with peak exposure of 45 µg/m^3^ for women pregnant during the mine fire [18]. No associations were found between PM2.5 exposure and birth weight or gestational age outcomes using routinely collected data from 763 mothers pregnant during the fire. However, among women with gestational diabetes, there was a 97 g increase in birth weight for every 10 µg/m^3^ increase in average PM2.5, and a 107 g increase for every 10 µg/m^3^ increase in peak PM2.5 exposure [18]. Exposure in utero was not associated with general practitioner (GP) attendance, or the use of asthma medications, steroid skin creams or antibiotics in infancy [19]. However, there was a significantly increased risk of parent-reported runny nose or cough (relative risk (RR) 1.09, 95% CI 1.02, 1.17) and doctor diagnosed upper respiratory tract infection or cold or flu (RR 1.35, 95% 1.14, 1.60) two to four years after the fire, for each 10 µg/m^3^ increase in average PM2.5 exposure among 79 children with intrauterine exposure [20]. Exposure to acute and prolonged fire smoke events are likely to have health effects on children at least in the short to medium term. More work will be needed to investigate long term health impacts in this population.

### 1.2. Effects of Bushfire Smoke Exposure in Children on Respiratory Health Outcomes

There are few publications reporting the effects of bushfire smoke exposure on healthcare utilisation in children, and no study has examined health outcomes for children whose exposure occurred in utero. A study in Colorado found significantly increased odds for ED presentation or hospitalisation for asthma in children associated with wildfire exposure (for each 1 µg/m^3^ increase in fire associated PM2.5 exposure, odds ratio [OR] 1.075, 95% CI 1.035, 1.115) [21]. A study of >97,000 ED visits in 2017 found that children exposed to the Californian Lilac Fire in 2017 (10-day exposure) had an excess of 16 (95% CI 11, 21) respiratory ED visits per day in all paediatric age groups, highest among children aged 0 to 5 years [22]. Another Californian study compared potential health effects and mechanisms such as alterations of T-cell populations and methylation patterns, after exposure to wildfires (36 children, September 2015) or controlled burns (32 children, March 2015) in the same region [23]. Air pollutant levels, including PM2.5, PM10, carbon monoxide (CO), nitrogen dioxide (NO_2_), nitrogen oxides (NO_x_), polyaromatic hydrocarbons, and elemental carbon, were all significantly higher during the wildfire compared to the controlled burn. Although this was a small study, there were trends towards adverse health effects, including wheezing episodes in children without prior asthma, and more exacerbations among children with asthma. After 90 days, children exposed to wildfire had a significantly lower percentage of type 1 T helper (Th1) cells in blood compared to children exposed to controlled burns [23]. In animal studies, monkeys exposed to wildfire smoke in infancy showed immune dysregulation (reduced induction of IL-8 protein following lipopolysaccharide [LPS] stimulation of peripheral blood mononuclear cells [PBMCs] in females and reduced interleukin [IL]-6 protein induction in males), as well as reduced lung function in adolescence (3 years of age) [24], suggesting that early life exposure to wildfires may have long term consequences for respiratory health.

Although no investigations to date have specifically addressed the relationship between bushfire smoke exposure and infant respiratory outcomes following in utero exposure, the Hazelwood study [19] examined associations between exposure to PM2.5 from coal fire emissions and infant outcomes in the general population. The authors detected an increased risk of antibiotic use in the first year of life (adjusted incidence rate ratio (IRR) 1.24, 95% CI 1.02, 1.50) with increased average PM2.5 exposure during infancy [19]. In addition, early childhood exposure (first 2 years of life) was associated with an increase in asthma inhaler use (RR 1.26, 95% CI 1.01, 1.58 per 100 µg/m^3^ peak PM2.5) two to four years later [20]. Among 84 children exposed postnatally, there was a significant worsening of lung function three years after the fire (average age 4.3 years), measured by the area under the reactance curve using the forced oscillation technique (FOT) [25].

### 1.3. Effects of Prenatal Ambient Air Pollution Exposure in Children on Perinatal and Respiratory Health Outcomes

There is a large body of literature examining adverse health effects in children following prenatal traffic-related air pollution exposure, including adverse perinatal outcomes [26] such as preterm birth [27], alterations in lung function at 5 weeks [1] and 6 years of age [28], and increased risk of asthma development [29]. Studies of air pollution exposure during pregnancy demonstrating adverse perinatal outcomes indicate that the fetus is highly susceptible to various environmental pollutants via in utero exposure [30]. A meta-analysis of 32 studies of PM2.5 exposure during pregnancy found the late stages of pregnancy had the highest magnitude associations with low birth weight, with a 15.9 g reduction in birth weight per 10 µg/m^3^ increment in PM2.5 exposure over the entire pregnancy [31]. In another systematic review and meta-analysis, birth weight was negatively associated with both PM10 and PM2.5; for every 10 µg/m^3^ increase in PM2.5, birth weight decreased by 22.17 g after adjusting for maternal smoking [26].

A study in Poland investigated prenatal exposure to outdoor and indoor air pollution and health effects in offspring up to 7 years of age [32]. There was a dose-response relationship between prenatal PM2.5 and the probability of recurrent lung infections, with respiratory problems increasing at PM2.5 levels as low as 20 µg/m^3^. Children with asthma were twice as likely to have episodes of bronchitis or pneumonia than children without asthma, and the probability of infection in relation to prenatal PM2.5 increased linearly in both groups, but more steeply in children with asthma.

### 1.4. Effects of Prenatal Stress on Children’s Respiratory Health

Prenatal stress is associated with adverse health consequences in childhood, including persistent wheeze up to age 5 years [7], and asthma development [33]. Some studies have also indicated that exposure to both prenatal stress (via negative life events) and prenatal air pollution (PM2.5) results in an even greater risk for childhood respiratory disease than either exposure alone [34], yet mechanisms such as endocrine effects and/or epigenetic changes remain unclear. Prenatal stress has been studied in the context of other natural disasters, such as the 2016 Fort McMurray wildfires [35] and the 2011 Queensland floods [36]. In the Queensland flood study, the Impacts of Events Scale Revised (IES-R) was used as a measure of subjective stress and the range of scores reported (up to 45) indicated some women experienced clinically significant post-traumatic stress disorder [36], which in some cases would be associated with suppression of the immune system [37]. An intervention to target prenatal maternal stress is already being tested in the context of several Canadian studies of prenatal exposure to natural disasters (Project Ice Storm in Montreal, Fort McMurray studies of wildfires) [38].

The 2019/2020 Australian bushfire crisis is known to have elicited stress among pregnant women with asthma (Asthma Australia, personal communication), and previous studies have associated prenatal stress with low birth weight and preterm birth [39], including a study linking in utero stress related to a natural disaster (Montreal Ice storm) with lowered birth weights [40]. Furthermore, low birth weight and preterm birth are known risk factors for asthma and respiratory illnesses in childhood. While there are some studies showing again that epigenetic changes may mediate these associations in the context of maternal stress [41], the influence of bushfire smoke exposure on maternal stress, epigenetic changes, and low birth weight have not been investigated.

Holstius et al. [11] proposed two potential mechanistic pathways linking wildfire exposure during pregnancy to lower birth weight; the first involved biological mechanisms resulting from exposure to the air pollution related to the fire smoke, while the second involved psychosocial mechanisms due to stress resulting indirectly or directly from the fires themselves. Living through a bushfire event may induce stress due to experiencing extreme heat, poor air quality, fear, threat, and/or loss [10].

## 2. DNAm as a Potential Epigenetic Mechanism Linking Smoke Exposure or Prenatal Stress to Childhood Respiratory Health Outcomes

DNAm is a common epigenetic mechanism, involving the addition of a methyl group to a DNA molecule, thereby influencing the expression and the splicing of genes. Most DNAm occurs at CpG sites, DNA sequences where a cytosine (C) immediately precedes a guanine (G) nucleotide.

Only one study has examined DNAm after exposure to wildfire smoke. This study focussed on children, whose immune systems were still developing, and who had reduced lung size. In 36 children from Fresno, California, higher *FOXP3* methylation was detected after wildfire exposure, compared to the 32 children with exposure to controlled burns [23], consistent with prior studies of the effect of air pollution on DNAm in children. These prior studies found that differentially methylated regions (DMRs) in the *FOXP3* promoter and *IL10* genes were different in children with asthma, and in the case of *FOXP3* methylation, also associated with exposure to ambient air pollution [42].

There are a limited number of investigations of newborn epigenetic markers of prenatal stress exposure, and no studies have been identified which examined epigenetic changes in the context of prenatal exposure to wildfires. Project Ice Storm studied 224 women who were pregnant or became pregnant within 3 months of an ice storm in Montreal in 1998. In a follow-up of 36 children exposed in utero, DNAm was examined in T-cells at age 13.5 years [43]. More than 1600 CpGs were identified to be associated with objective maternal stress, with the majority from genes related to immune function. A systematic review examining epigenetic mechanisms linking prenatal stress (measured as depression, anxiety, or presence of stressful life events) to fetal hypothalamic pituitary adrenal (HPA) axis function, described 12 studies examining DNAm in the glucocorticoid receptor gene nuclear receptor subfamily 3 group C member 1 (*NR3C1*) in cord blood, placenta, or buccal cells in the first 5 months of life [44]. All studies examined exon 1_F_, and two studies found an association between 3rd trimester anxiety and newborn *NR3C1* hypermethylation at CpG2 within exon 1_F_ [45,46]. Hence, the methylation of *NR3C1* is considered an important biomarker of environmental stressors.

Other investigations which may be informative for future work on bushfire smoke exposure are those examining tobacco smoke exposure in pregnancy and changes in DNAm in umbilical cord blood. Multiple studies from different countries participated in an epigenome-wide meta-analysis indicating the differential DNA-methylation (DNAm) of about 600 genes [47,48]. In 2015, a small study of 20 mothers and newborns showed statistically significant differences in 31 CpG sites associated with 25 genes, associated with tobacco smoke exposure in utero, in the absence of fetal growth restriction [49]. In the Avon Longitudinal Study of Parents and Children (ALSPAC) cohort, among 800 mother–offspring pairs, changes in cord blood methylation at 15 CpGs sites was associated with smoking during pregnancy, with a dose-dependent response noted. Some of these changes persisted at 7 years and 17 years of age in offspring [50]. A study from the Generation R cohort found that changes in cord blood DNAm at 1391 CpGs was related to continued maternal smoking during pregnancy, whereas quitting smoking early in pregnancy and paternal smoking were not associated with DNAm changes at 5915 CpGs known to be related to maternal smoking [51].

A recent study suggests that changes in DNAm (specifically lower methylation at CpG locus cg05575921 on the *AHRR* gene) may mediate the association between tobacco smoke exposure in pregnancy, and asthma in childhood [52]. The Aryl hydrocarbon receptor repressor (*AHRR*) gene mediates metabolism of the toxic components of cigarette smoke. Exposure to polycyclic aromatic hydrocarbons (PAH, found in both in cigarette smoke and PM2.5) has been linked to variations in *AHRR* methylation. Another study has investigated methylation at cg05575921 on the *AHRR* gene in relation to air pollution, and specifically PM2.5 concentrations, among 708 non-smoking adults in Taiwan [53]. In blood samples, methylation decreased as PM2.5 in the area of residence increased. A one-unit increase in PM2.5 was associated with 0.00115 (*p* < 0.001) lower cg05575921 methylation levels. Multiple international investigations have shown that air pollution can result in differential DNA-methylation [54,55,56,57]. However, it remains unclear whether air pollution data can be generalized for the health effects of bushfires, given the different chemical composition of these two sources of pollutants.

A recent study of prenatal exposure to PM10, PM2.5, and PM1 in China found that the risk of preterm birth was positively associated with both PM2.5 and PM1 from 12 to 20 weeks gestation, and that cord blood long interspersed nucleotide element 1 (*LINE-1*) methylation was negatively correlated with all PMs [27]. Decreased *LINE-1* methylation may also have a role in triggering inflammatory responses, a potential mechanism involved in preterm birth [27]. Another Chinese air pollution study showed that prenatal exposure to PM10 was associated with DNAm of the superoxide dismutase 2 (*SOD2*) promoter in both maternal and cord blood [58]. In addition, placental and cord blood changes in DNAm, in genes related to both inflammation and alterations in metabolism, were found to be related to prenatal exposure to NO_2_ and ozone [59]. The authors suggest that specific changes to DNAm may be important biomarkers of prenatal exposure, as well as mediators of adverse health outcomes associated with air pollution.

A recent study of two population-based cohorts showed differential blood DNAm of 8 CpG sites in 8 genes related to development and lung morphogenesis at birth, which were associated with lung function trajectories over the first 26 years of life [60]. This was the first study to examine health trajectories across a significant period of time, specifically at ages 10, 18, and 26 years in the Isle of Wight birth cohort, and ages 8, 15, and 24 years in the ALSPAC cohort. This suggests the importance of epigenetic changes in early life contributing to later susceptibility to lung disease and highlights why further research into epigenetic changes in response to bushfire smoke exposure in utero is needed. Thus, further work is required to determine which exposures that occur in utero determine these patterns of DNAm at birth.

## 3. Conclusions

Exposure to bushfire smoke, prenatal stress from natural disasters, and/or air pollution has effects on pregnancy and birth outcomes and is an area of significant public health importance and interest, with climate changes increasingly on the global agenda, and the occurrence of bushfires rising globally. Pregnant women are a vulnerable population, and the specific effects of bushfire smoke exposure on infant outcomes, such as development of asthma are unknown. While there is abundant literature on the effects of ambient air pollution during pregnancy, there is relatively little related to bushfire smoke specifically, with some important gaps. This review has covered the literature and proposed epigenetics as a potential mechanism linking exposure to PM2.5 and other components of fire emissions and/or maternal stress from bushfire events to adverse perinatal and respiratory outcomes in the offspring (Figure 1). Changes in DNAm at CpG sites in genes which have been previously shown to have associations with exposures to wildfire smoke (in children), air pollutants (in children and in utero), tobacco smoke (in utero), and prenatal stress may be involved.

Future research in large pregnancy cohorts which can determine the early life health effects in children exposed to bushfire smoke in utero, particularly in children at high risk of developing wheezing, bronchiolitis, and asthma, are needed. If effects on offspring are identified, further studies to minimise or mitigate in utero exposure can be undertaken, in particular regarding stress, or air pollution-related effects. The identification of epigenetic markers could initiate novel methods to identify exposed offspring at higher risk of adverse health effects, and future trials to reduce their risks. If we can partition exposure risks, then interventions could be targeted. Hence, there is a need to distinguish between the effect of maternal stress and maternal exposure to toxic substances. For example, specific interventions aimed at managing prenatal stress will be very different to those that target PM2.5 exposure. Information on epigenetic pathways will allow focused public health and medical mitigation in more susceptible children. On the other hand, if studies do not systematically identify adverse health effects, then this would provide reassuring information to mothers, health care professionals, and the community regarding the as-yet unknown delayed risks of intense bushfire smoke exposure during pregnancy on outcomes in the offspring.

The American Thoracic Society recently published a workshop report which highlighted that long-term health effects of repeated smoke exposures across fire seasons are unknown and require more research [61]. In addition, there has been little research on the efficacy of public health interventions designed to mitigate exposure. With millions of people in the United States exposed to bushfires, often more than once during a fire season, there is a need for research on repeated exposures across fire seasons [61]. While this is not as relevant in the context of the relatively short time frame of pregnancy, for infants who were exposed both in utero and in early life, there may be additional associations with their future health which are yet to be investigated.

New research is needed to determine early life health effects in children and to investigate the biological mechanisms involved. The evaluation of bushfire effects during pregnancy on children’s health is of increasing and urgent global significance, and there is a need for global studies to provide a strong evidence base for public health decision making.

## Figures and Tables

**Figure 1 ijerph-18-07465-f001:**
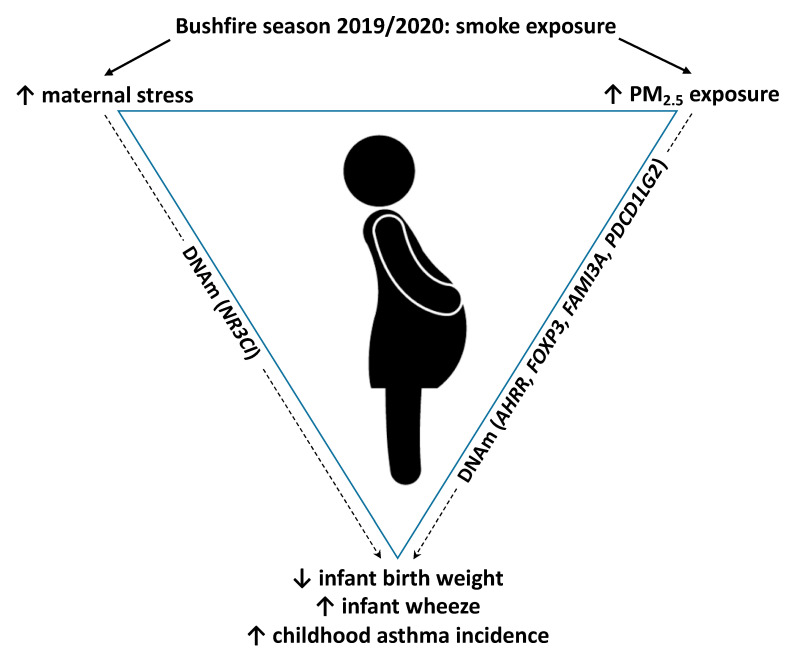
Proposed associations between exposure to stress and PM2.5 during the bushfire season, and adverse infant outcomes, mediated by epigenetic mechanisms.

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
