# Peer review of "Exposure to Stress and Air Pollution from Bushfires during Pregnancy: Could Epigenetic Changes Explain Effects on the Offspring?"

_ijerph, 2021, doi:10.3390/ijerph18147465_

Round 1

Reviewer 1 Report

This article is a comprehensive review on the effect of prenatal stress (due to experiencing natural disasters) and/or exposure to air pollutants (bushfire smoke and other sources of air pollution) on perinatal health of the offspring. The health effects include preterm birth, low birth weight, and susceptibility to respiratory diseases such as asthma. Specifically, they state based on research existing in the literature that such exposure may affect the epigenome, particularly via altering the DNA methylation profile of the affected individuals, proposing epigenetics as a potential link between the exposure to air pollutants/stress and health consequences for the newborns.

The paper is organized and well-prepared. I believe not only it can attract biologists and physicians, but the general public, due to the increasing global attention towards the climate change issue.

Major:

The title should be more general, since the authors do not talk only about exposure to bushfire smoke. The fist paragraph of the conclusion should also be adapted accordingly, with a more balanced emphasize on stress and air pollution. 

Minor:

  • Page 2, last paragraph, line 1: “effects” has been typed twice.
  • Page 2, last paragraph, line 12: Too much space between “2.5” and “m” and many other places between the quantity and the unit. On the other hand, there is no space between the value and the unit at some other parts of the text, such as Page 3, paragraph 3, lines 6 and 7. Either of these formats should be selected and remain consistent throughout the text.
  • Dates format (for example, “1st October” on Page 2, last paragraph, line 14) seems unusual. “October 1st” is a more common style.
  • Page 3, paragraph 3, line 5, “from” is redundant.
  • Page 3, paragraph 3, last sentence, it seems that what is meant by “normally” is “under normal conditions”.
  • Page 4, paragraph 1, line 4, there seems to be an extra space between “studies” and “focused”. Also “focused” is spelled wrong.
  • There are several places where present tense is used, while always past tense should be used, especially since this is a review paper and the utilization of present tense implies that the authors are referring to the current study. For instance, on Page 4, paragraph 3, lines 3-7: “This fire burned for 51 days in February and March 2014, covering towns in regional Victoria in plumes of smoke and ash. In this study, 571 children who were exposed either in utero, up to 2 years of age, or conceived after the fires, are being followed-up, while de-identified data from 3679 children born between March 2012 and December 2015 is also being used to assess effects on child health outcomes.”
  • Page 6, second to last paragraph of 1.4, last line, “has” should be replaced with “have”.
  • Page 6, Section 2, paragraph 3, first few lines, the statement “none have been identified in the context of wildfires” sounds confusing, since it has already been mentioned in the previous paragraph that “one study has examined DNA methylation after exposure to wildfire smoke”. In the latter paragraph, do the authors mean that there was no study that considered prenatal stress exposure that was a result of experiencing a wildfire event?
  • Page 6, last paragraph, line 4, “a” should be replaced with “an”. In the next line, the references should be moved to the left of the period.
  • The abbreviation “DNAm” has been introduced twice. It should only be stated the first time. Also, there is inconsistency in using the abbreviation versus the complete term in several places. I suggest using the abbreviation throughout the manuscript to make the best use out of its introduction.
  • Page 7, paragraph 4, line 8, “epigenetics changes” should be replaced with “epigenetic changes”.

Reviewer 2 Report

1.- All the abbreviations must be defined: PM2.5, PM10, NO2, NOX, CO, LINE-1

2.- The following lowing paragraph is confusing: "In addition, changes in DNA methylation were found to be related to prenatal exposure to NOand ozone in placenta and cord blood, in genes related to inflammation and alterations in metabolism": it is not clear wether the exposure was in placenta and cord blood nor whether the methylation was in genes related to inflammation and alterations in metabolism. Please rewrite 

3.- Considering that only few genes have been associated with methylation patterns in pollution and wildfires, it would be interesting to know if there are studies addressing all the epigenomic changes or modifications in these conditions. 
